# Catalytic asymmetric nucleophilic fluorination using BF₃·Et₂O as fluorine source and activating reagent

Weiwei Zhu[1], Xiang Zhen[2], Jingyuan Wu[1], Yaping Cheng[1], Junkai An[2], Xingyu Ma[1], Jikun Liu[2], Yuji Qin[1], Hao Zhu[2], Jijun Xue[2] & Xianxing Jiang [1✉]

Fluorination using chiral catalytic methods could result in a direct access to asymmetric fluorine chemistry. However, challenges in catalytic asymmetric fluorinations, especially the longstanding stereochemical challenges existed in BF₃·Et₂O-based fluorinations, have not yet been addressed. Here we report the catalytic asymmetric nucleophilic fluorination using BF₃·Et₂O as the fluorine reagent in the presence of chiral iodine catalyst. Various chiral fluorinated oxazine products were obtained with good to excellent enantioselectivities (up to >99% ee) and diastereoselectivities (up to >20:1 dr). Control experiments (the desired fluoro-oxazines could not be obtained when Py·HF or Et₃N·3HF were employed as the fluorine source) indicated that BF₃·Et₂O acted not only as a fluorine reagent but also as the activating reagent for activation of iodosylbenzene.

[1] Guangdong Provincial Key Laboratory of Chiral Molecule and Drug Discovery, School of Pharmaceutical Sciences, Sun Yat-Sen University, Guangzhou, China. [2] State Key Laboratory of Applied Organic Chemistry, College of Chemistry and Chemical Engineering, Lanzhou University, Lanzhou, Gansu, China. ✉email: jiangxx5@mail.sysu.edu.cn

**B**eing called as "a small atom with a big ego", fluorine acts as a significant and increasingly important role in the fields of organic chemistry, pharmaceuticals, agrochemicals and material chemistry[1–4]. The fluorinated molecules often display higher thermal and metabolic stabilities, lower polarity, and weaker intermolecular interactions due to the strong C−F bond and unique properties of F atom[5]. Therefore, these unique properties of fluorine-contained compounds make the development of efficient strategies, especially of catalytic asymmetric reaction for fluorination of molecules as one of the hottest areas in organic synthesis[2,3]. Nevertheless, the asymmetric fluorine organic chemistry still represents a considerable challenge to date[6]. In the wake of the emergence of the first electrophilic enantioselective fluorination of enolates using chiral *N*-fluoro camphorsultam reagent reported by the group of Lang[7], significant progress for enantioselective fluorination studies have been presented[4,8,9] because of contributing to the development of catalytic asymmetric methodologies for electrophilic fluorine reagents (F+ reagents), such as *N*-fluorobenzenesulfonimide (NFSI)[10–12], *N*-fluoropyridinium salts[3], and Selectfluor (Fig. 1a)[13–18]. These reagents exhibited efficient transfer of fluorine atom under the asymmetric fluorination, however, their industrial applications were significantly restricted by the poor atom economy in fluorination, expensive synthesis and other inherent characteristics of electrophilic reagent. Alternatively, nucleophilic fluorine reagents (F− reagents) have been attracting considerable interest recently since the relative stability and low-cost. Considerable advances have recently been achieved in this field involving catalytic asymmetric fluorination of keto esters[19,20] and alkenes[21–25] employing pyridine·HF as a fluorine reagent, catalytic asymmetric fluorination of allylic trichloroacetimidates using a combination of Et₃N·HF with Iridium complex[26], asymmetric ring-opening fluorinations of meso-epoxides (aziranes) using PhCOF, HF-reagents or AgF as the fluorine source in metal-catalyzed system[27–29], and other asymmetric transformations in the presence of metal fluorides (KF, CsF or AgF)[30–32] (Fig. 1b). Despite these elegant works, several practical disadvantages still discouraged their further large-scale

utilization in the area: the high toxicity and biohazard for HF-bases, and metal fluorides poor solubility in organic solvents coupled with limited strategies to control reactivity.

Ideally, one low-toxic, stable and commercially cheap available nucleophilic fluorine reagent would drastically promote enantioselective fluorine synthetic innovation and industrial development. As a versatile Lewis acid, commercially available BF₃·Et₂O is easy to prepare and is widely being used in various organic transformations. As early as 1960, it was discovered that as a nucleophilic fluorine reagent could be applied in the fluorination of ring opening of mesoepoxides[33]. The development of BF₃·Et₂O mediated reactions in half a century reveals that the BF₃·Et₂O can also be applied to fluorinations of alkenes[34–36], alkynes[37], arenes[38,39] and other fluorine organic chemistry[40–43]. Although these efficient achiral methodologies have been well-established, to date, the longstanding stereochemical challenges of the BF₃·Et₂O-based fluorination have not yet been addressed, probably impeded by several hurdles: intense competition for the role of BF₃ between a nucleophilic fluorine reagent and a Lewis acid, the difficulty in achieving stereocontrol of fluorine atom, the competition from the uncatalyzed background reaction and other side reactions. Undoubtedly, the advent of enantioselective approach is long overdue that would be welcome.

In terms of the operational and environmental advantages associated with organocatalysis, we speculated that a metal-free mild reaction system with a chiral iodine catalyst (CIC) might meet the aforementioned challenges[21,44,45]. Its activated oxidants forms could form the iodine (III) catalyst with a typical structure type of trigonal bipyramidal geometry, thus this type of robust organocatalyst has been commonly used for asymmetric nucleophilic addition reactions[44,45]. The unique stereoscopic configuration of iodine (III) and well-defined steric hindrance of the iodine (III) catalyst bearing chiral ligand can be readily applied, leading to complete stereo-control in fluorination of olefins using BF₃·Et₂O as a nucleophilic reagent. Rapid cyclization and its simultaneous BF₃·Et₂O nucleophilic fluorination are viable with a CIC to suppress the pure intramolecular cycloaddition and other side reactions.

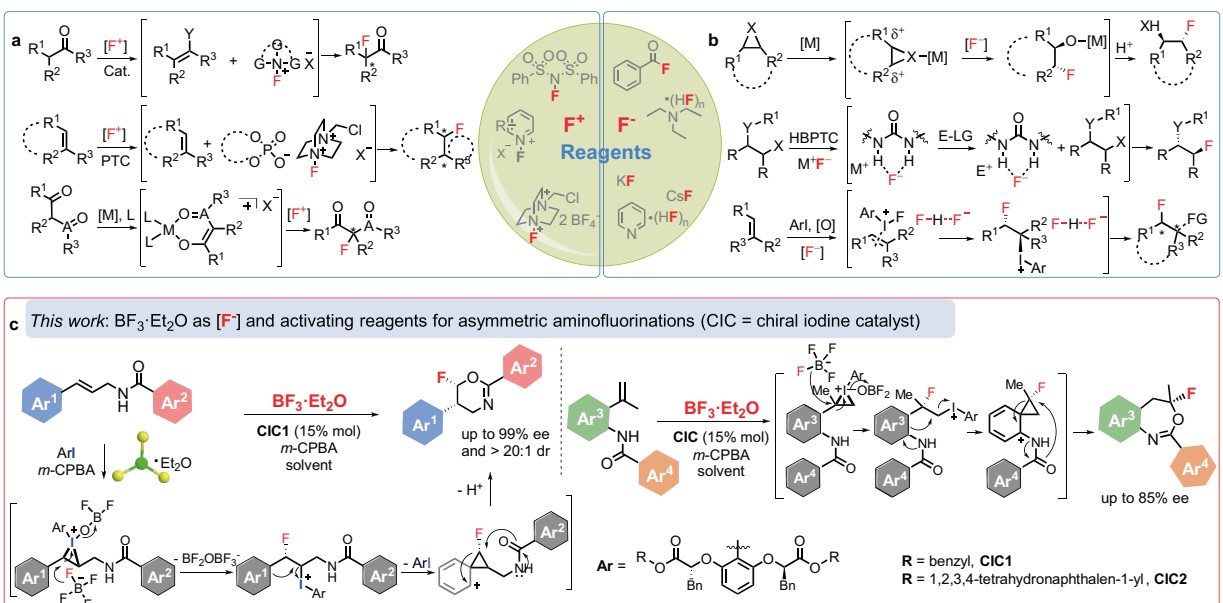

**Fig. 1 Examples of common fluorine reagents applied in asymmetric fluorinations. a** Asymmetric fluorinations using electrophilic fluorine (F+) reagents. **b** Asymmetric fluorinations using nucleophilic fluorine (F−) reagents. **c** The first example of using BF₃·Et₂O as F− Reagent for asymmetric fluorinations (this work). [F+], electrophilic fluorine reagents; [F−], nucleophilic fluorine reagents; PTC, phase-transfer catalyst; [M], metal catalyst; L, ligand; HBPTC, hydrogen bonding phase-transfer catalysis; [O], oxidant; *m*-CPBA, *Meta*-Chloroperoxybenzoic acid.

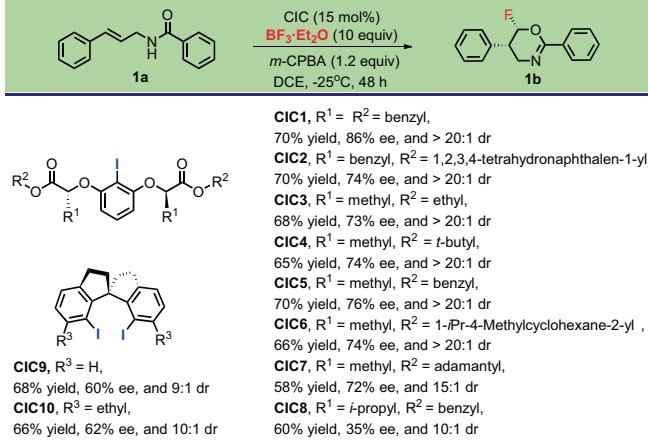

**Fig. 2 Optimizing CIC structure in the model reaction.** Notations for CICs: the reaction of **1a** (0.1 mmol), BF$_3$·Et$_2$O (1.0 mmol), m-CPBA (0.12 mmol) and catalyst (15% mol) was carried out in 4.0 ml of 1,2-dichloroethane (DCE) at −25 °C for 48 h; isolated yields are reported. The ee values were determined by chiral HPLC analysis and the dr values were determined by $^{19}$F NMR.

Herein, we introduce a catalytic asymmetric fluorination strategy that involves BF$_3$·Et$_2$O as nucleophilic fluorine source with chiral iodine catalysts. Various F-contained products were obtained with good to excellent diastereoselectivities (>20:1 dr) and enantioselectivities (up to >99% ee) (Fig. 1c).

## Results and discussion

**Catalyst optimization.** After several initial trials (Supplementary Table 1), we set out to optimize the model catalytic asymmetric aminofluorination of N-cinnamylbenzamide (**1a**) in the presence of 15 mol% of ligand loading using BF$_3$·Et$_2$O as the fluorine reagent in DCE at −25 °C with m-CPBA as an oxidate (Fig. 2). The significant difference in stereoselectivity was observed under these reaction conditions, whereas good yields were afforded with the addition of structure diverse chiral iodine catalyst CICs. These results suggested that the substituents of the catalysts have a strong influence on the stereochemistry of the reaction. In general, compared to spiro-CICs, the linear CICs could give the higher stereoselectivity. The **CIC1** was proved to be the best catalyst, providing the desired chiral fluorinated oxazine **1b** with excellent diastereoselectivity (>20:1 dr) and enantioselectivity (86% ee) in 70% yield.

**Asymmetric aminofluorination of N-cinnamylbenzamides using BF$_3$·Et$_2$O.** Results of experiments under the optimized conditions that probe the scope of the reaction are summarized in Fig. 3. Substrate scope of the reaction was investigated with a variety of substituted N-cinnamylbenzamides under the optimal reaction conditions. As shown in Fig. 3a, variation of the electronic properties of substituents at either Ar$^1$ or Ar$^2$ of N-cinnamylbenzamides with different steric parameters were tolerated, affording the desired products with good to excellent diastereoselectivities (2:1–>20:1 dr) and enantioselectivities (80–>99% ee) in good yields (45–75%). Gratifyingly, the fluorinated oxazine products bearing Ar$^2$ with high steric hindrance were still obtained in good yields and excellent stereoselectivities (85–>99% ee, **15b–24b**). The naphthyl-substituted N-cinnamylbenzamides could also be tolerated, and gave the corresponding fluorinated products (**26b**, **27b** and **31b**) in excellent diastereoselectivities (up to >20:1dr) and enantioselectivities (86–87% ee) with good yields. Furthermore, as expected, the catalytic system also proved to be efficient for the N-cinnamylbenzamides with heterocycles or

"complex substituents" on Ar$^2$ ring (Fig. 3b, **35b–42b**), again leading to good yields (45–72%) in high to excellent diastereoselectivities (10:1–>20:1dr) and enantioselectivities (80–99% ee). It is worth noting that the catalytic asymmetric aminofluorination of complex natural product structures could also be achieved efficiently in high to excellent diastereoselectivities and enantioselectivities.

Additionally, the gram-scale experiment was conducted to evaluate the applicability of our asymmetric fluorination method by using **4a** (Fig. 3c), the desired product was obtained with excellent diastereoselectivity (>20:1 dr) and enantioselectivity (92% ee). The result suggested that our protocol was promising in future industrial applications. It was interesting that compared to the PTC catalyzed process[46], different fluorinated products could be obtained from the same substrates using current process (indicating a different catalytic progress). The relative and absolute configurations of the products were determined by X-ray crystal structure analysis of **4b** (see the Supplementary Information).

**Asymmetric aminofluorination of N-(2-(prop-1-en-2-yl)phenyl)benzamides.** To further understanding of the scope of this catalytic system, substrates **43a–54a** were employed to undergo the fluorination process. To our delight, various substituted N-(2-(prop-1-en-2-yl)phenyl)benzamides including either electron-donating substituents or steric hindrance substituents at different positions on the Ar$^4$ ring, as well as 3,5-ditrifluoromethyl substituents could be tolerated, affording the corresponding fluorinated products (**43b–54b**) with high enantioselectivities (80–85% ee) and isolated yields (80–88% yield) (Fig. 4). These substrate scope expanding experiments showed the hypothesis to introduce the combination of BF$_3$·Et$_2$O and CIC into direct, catalytic asymmetric fluorinations can be achieved. This method expanded the structures of the fluorinated products and provided a benign access to asymmetric nucleophilic fluorinations.

**Mechanism studies.** On the basis of the experimental results described above, we have proposed a possible mechanism to explain the stereochemistry of the catalytic asymmetric nucleophilic fluorinations (Fig. 5). To gain a better understanding of the process of this catalytic fluorination system, we also conducted control experiments (Fig. 5a) and density functional theory (DFT) calculations (Fig. 6). It is worth noting that we could not obtain the desired fluorinated products when we used or Py·HF or Et$_3$N·3HF instead of BF$_3$·Et$_2$O (Fig. 5a, equation 1). When PhIF$_2$[47] was applied as the hypervalent iodine reagent and fluorine source, we didn't obtain the **1b** as well (Fig. 5a, equation 1). Thus, these results indicate that BF$_3$·Et$_2$O acted not only as the nucleophilic fluorine source, but also as the activating reagents for activation of iodosylbenzene (**Int1**). which is distinct from previously catalytic nucleophilic fluorination process reported by Jacobsen's group[21]. Based on previous work[29,35,48,49], control experiments and DFT calculations, the plausible catalytic cycle was shown in Fig. 5b, and associated free energy profile was shown in Fig. 6.

At first, the aryl iodine catalyst is oxidized by m-CPBA to form Ar−I=O (**Int1**), and **Int1** is found to be 2.8 kcal/mol lower in energy than ArI. (Fig. 6). Then **Int2** is formed through the activation of Ar−I=O by BF$_3$·Et$_2$O[44], the energy of this intermediate is calculated to be 23.8 kcal/mol lower than **Int1** (Fig. 6). The electrophilic addition of iodine (III) to the double bond of **1a** forms **Int3**[34], during which the energy barrier is found to be 3.6 kcal/mol. Then with the assistance of BF$_3$, **Int3** releases anionic [BF$_4$]$^-$ to form **Int3$^+$**. The nucleophilic attack of [BF$_4$]$^−$ (generated in previous step) on **Int3$^+$** at C1 position afford the **TS1**[43], the energy barrier of this step is 13.1 kcal/mol. There were

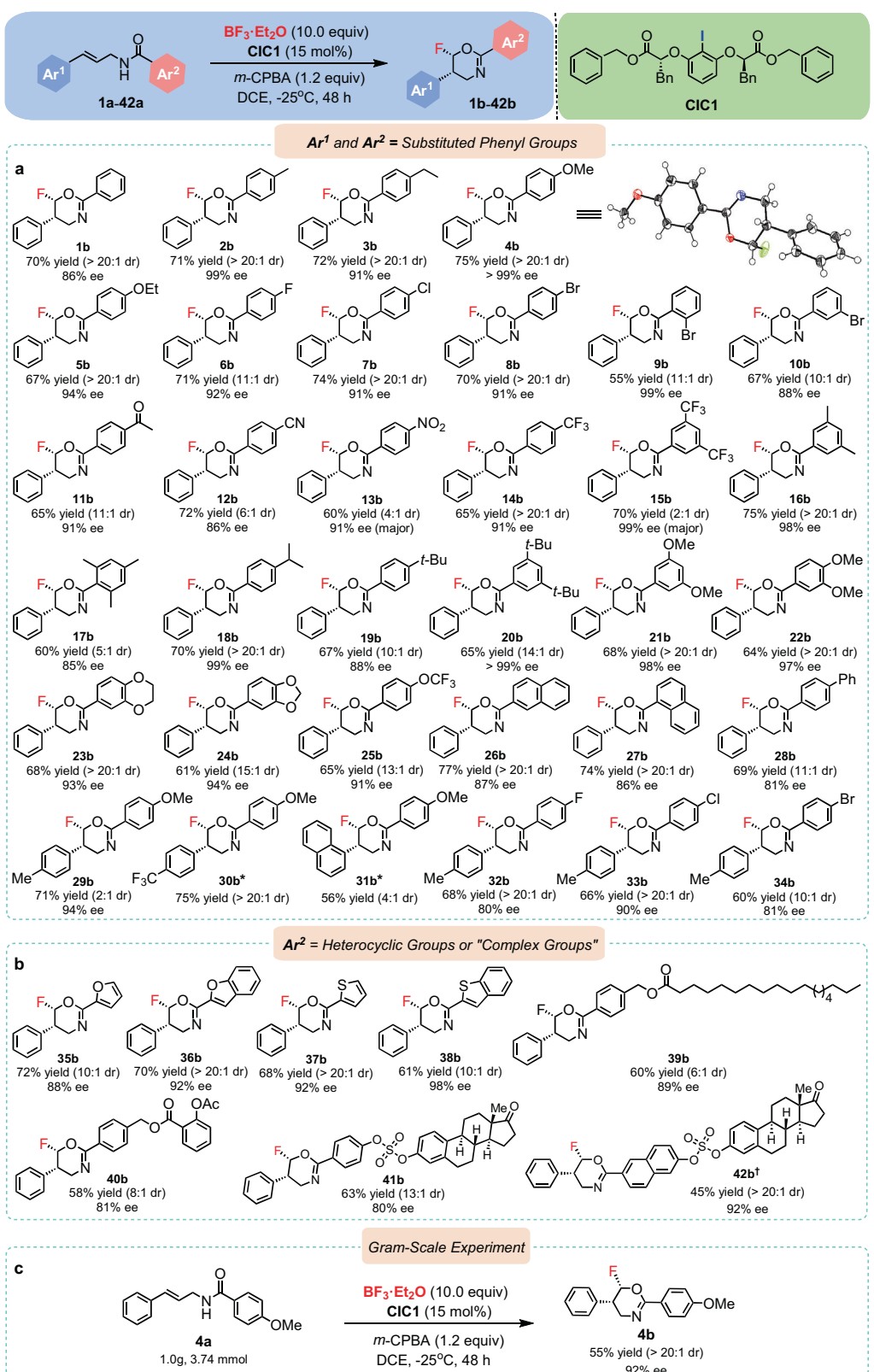

**Fig. 3 Substrate scope of the nucleophilic asymmetric fluorinations.** Reactions were conducted on a 0.2 mmol scale with 10.0 equivalents of BF$_3$·Et$_2$O in DCE (8.0 mL) at −25 °C for 48 h. The absolute configuration of **4b** was assigned by X-ray crystallography (structure shown), and the configuration of all other products was assigned by analogy. **a** Substrate scope of *N*-cinnamylbenzamides bearing various substituents on Ar$^1$ and Ar$^2$ ring. **b** Substrate scope of *N*-cinnamylbenzamides with "complex substituents" on Ar$^2$ ring and Ar$^2$ = heterocyclic groups. **c** Gram-scale reaction with **4a**, **CIC1**, *m*-CPBA and BF$_3$·Et$_2$O. Isolated yields are reported. *The ee values of **30b** and **31b** could not be detected by HPLC. †**CIC2** was employed as the catalyst for **42b**. The ee values were determined by chiral HPLC analysis and the dr values were determined by $^{19}$F NMR. *t*-Bu, *tert*-butyl; Ph, phenyl; Me, methyl; Ac, acetyl.

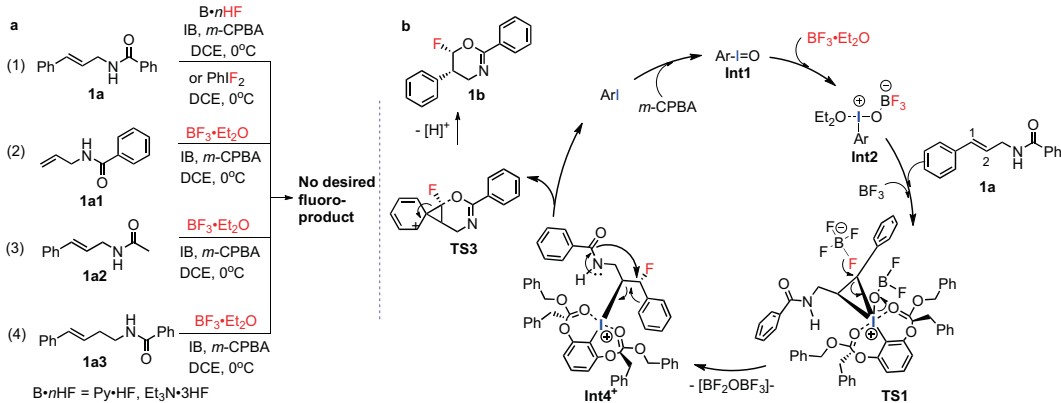

**Fig. 4 Substrate scope expanding of the nucleophilic asymmetric fluorinations using BF$_3$·Et$_2$O as the fluorine source.** Reactions were conducted on a 0.2 mmol scale with 10.0 equivalents of BF$_3$·Et$_2$O in C$_6$H$_5$F (8.0 mL) at −42 °C for 20 h; isolated yields are reported. *CIC2 was applied as the catalyst for the formation of **53b** and **54b**. The ee values were determined by chiral HPLC analysis. Me, methyl; *t*-Bu, *tert*-butyl; Et, ethyl.

**Fig. 5 Plausible mechanism of the catalytic asymmetric nucleophilic fluorinations. a** Control experiments. **b** Plausible catalytic cycle.

intramolecular n−σ* interactions between the electron-deficient iodine (III) center and the carbonyl groups[50,51]. For the impacts of steric hindrance, the nucleophilic attack of F− on the *Si* face was favored (Fig. 5b), and this is in consistence with the experimental results and DFT calculations (Fig. 6b). As seen from the energy profile, the enantioselectivity is determined by the addition of F- with [BF$_4$]- as fluorine source to the cationic **Int3**+ (Fig. 6a). The energy of **TS1** versus *ent*-**TS1**, the transition state to the minor product, is compared. Surprisingly the two **TS**s exhibit huge energy difference of 14.9 kcal/mol (Fig. 6b). A closer observation on the geometry shows that *ent*-**TS1** is much later, and bears a significantly longer I-O distance. In both **TS**s, the hypervalent I(III) atom is stabilized by interaction with the amide oxygen atom in the alkene. It could be proposed that the relative direction of alkene and catalyst in *ent*-**TS1** disabled the feasible I-O interaction that provides essential stabilization to the iodane, leading to both a later transition state and much higher energy. The formation of **Int4** is achieved through the interaction between **TS1** and BF$_3$, and **Int4** is found to be 23.7 kcal/mol lower in energy than **TS1**. Then **Int4** releases anionic [BF$_2$OBF$_3$]− to form **Int4**+, and **Int4**+ is found to be 5.5 kcal/mol lower in energy than **Int4**. Dearomatization of Ar$^1$ ring of **Int4**+ by intramolecularly nucleophilic attack of the Ar$^1$ on *Si* face at *C*2 position (Fig. 5b) furnish the cyclopropyl compound **TS2**

(Fig. 5b)[49], which was calculated to be +11.1 kcal/mol in energy relative to **Int4**+. And then **Int5** is formed with 1.7 kcal/mol of energy barrier relative to **TS2**. The hypervalent iodine (III) **Int5** underwent reductive elimination to afford **Int6** with 6.8 kcal/mol of energy barrier. **TS3** was formed by the intramolecularly nucleophilic attack of the amide oxygen on *C*2 position with 4.5 kcal/mol of energy barrier relative to **Int6**. Here the nucleophilic attack of the amide oxygen takes place regioselectively at the higher substituted carbon atom of the cyclopropane unit[49]. Ring opening of the spirocyclopropyl ring **TS3** takes place intramolecularly via a cyclization with simultaneous ring expansion to the six-membered cationic **Int7** (Fig. 6)[49]. The calculated energy barrier for this step is −38.9 kcal/mol relative to **TS5**. Control experiment (Fig. 5a, equation 2) demonstrated the necessity of Ar$^1$ ring. Moreover, the aromatic ring (Ar$^2$) is essential to stabilize cationic **Int7** (Fig. 5a, equation 3). With the assistance of [BF$_2$OBF$_3$]− anion, **Int7** can be deprotonated to generate final product **1b** (Fig. 6a). Besides, lengthening of carbon chain could not result in a desired fluoro-product according to the control experiment (Fig. 5a, equation 4). In a word, the formation of fluorinated oxazines follows a fluorination/1, 2-aryl migration/cyclization cascade[49].

We have disclosed an efficient asymmetric fluorinations process that has enabled the development of the first highly enantioselective

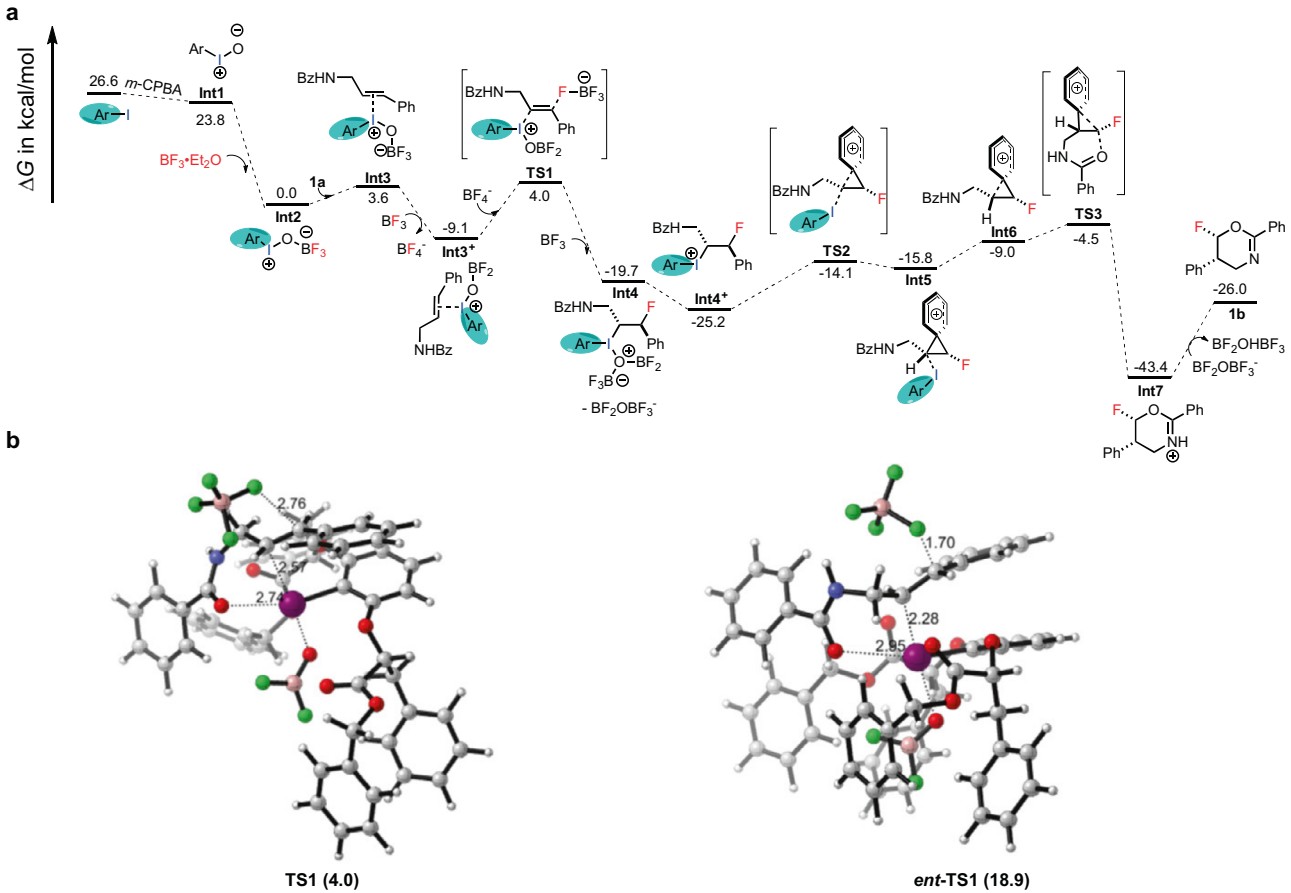

**Fig. 6 Calculated energy profile and DFT calculations for enantioselectivity of the catalytic asymmetric fluorination. a** DFT-computed reaction profile for the catalytic asymmetric fluorination of **1a** and BF₃·Et₂O in the presence of **ClC1** and *m*-CPBA. **b** DFT-optimized stereo-determining transition state (**TS1**) structures and their relative energies for **Int2**.

fluorination reaction (up to 99% ee and >20:1 dr) using BF₃·Et₂O as the fluorine source and dual-activating reagent. Moreover, the substrate expanding experiments further demonstrated the wide applicability of current method. This process provides not only a direct access to fluoro-oxazine/benzoxazepine skeletons, but also a foundation for further development of new types of asymmetric nucleophilic fluorinations in future applications. The studies of the applicability of this asymmetric fluorination methodology using other substrates are going on in our group.

## Methods

**General procedure for synthesis of 1b–42b.** The substrate (0.2 mmol) and catalyst (15 mol%) were mixed into the reaction tube, and then DCE (8.0 ml) was added. The mixture was cooled to −25 °C, after stirring for 10 min at this temperature, *m*-CPBA (1.2 equiv.) was added in one portion, followed by addition of BF₃·Et₂O (10.0 equiv.) dropwise. The reaction was run at −25 °C for 48 h. The reaction mixture was poured into saturated NaHCO₃ (aq), the organic layer was collected and washed with brine, dried over Na₂SO₄, and concentrated under reduced pressure in the presence of basic Al₂O₃, Column chromatography (basic Al₂O₃, 200–300 mesh, EtOAc-hexane (0.5% Et₃N) elution: hexane/EtOAc (V/V) = 50:1 ~ 5:1) gave the corresponding fluorinated products.

**General procedure for synthesis of 43b–54b.** The substrate (0.2 mmol) and catalyst (20 mol%) were mixed into the reaction tube, and then C₆H₅F (8.0 ml) was added. The mixture was cooled to −42 °C, after stirring for 5 min at this temperature, *m*-CPBA (1.2 equiv) was added in one portion, followed by addition of BF₃·Et₂O (10.0 equiv) dropwise. The reaction was run at −42 °C for 20 h. The reaction mixture was poured into saturated NaHCO₃ (aq) sulotion, the organic layer was collected and washed with brine, dried over Na₂SO₄, and concentrated under reduced pressure in the presence of basic Al₂O₃, Column chromatography (basic Al₂O₃, 200–300 mesh, EtOAc-Hexanes (0.5% Et₃N) elution: hexanes/EtOAc (V/V) = 100:1 ~ 25:1) gave the corresponding fluorinated products.

**DFT calculations**. All calculations were carried out with the Gaussian 09 software[52]. The B3LYP functional[53] was adopted for all calculations in combination with the D3BJ dispersion correction[54]. For geometry optimization and frequency calculations, the SDD ECP and basis set[55] was used for I and 6-31 G(d) for others[56,57]. The singlet point energy calculations were performed with a larger basis set combination, in which the def2-TZVP basis set[58] was used for I, and 6−311 + G (d,p)[59,60] for others. The SMD implicit solvation model[61] was used to account for the solvation effect of DCE when performing single point energy calculations.

**Reporting summary**. Further information on research design is available in the Nature Research Reporting Summary linked to this article.

## Data availability

All relevant data are available from the corresponding author upon reasonable request. All the data supporting the findings of this study are available within this article, and supplementary information files. The authors declare that all data generated or analyzed during this study are included in this Article (and its Supplementary Information). The X-ray crystallographic coordinates for the structure of **4b** are available free of charge from the Cambridge Crystallographic Data Centre under deposition number CCDC 1960281.

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

## Acknowledgements
We appreciate the financial support from the National Natural Science Foundation of China (No. 91853106), the Program for Guangdong Introducing Innovative and Enterpre-neurial Teams (No. 2016ZT06Y337), Guangdong Provincial Key Laboratory of Construction Foundation (No. 2019B030301005), Shenzhen Science and Technology Program (JSGG20200225153121723), and the Fundamental Research Funds for the Central Universities (No. 19ykzd25). We also thank Dr. Chaoxian Yan for his help in DFT calculations.

## Author contributions
X.X.J. and W.W.Z. conceived of and directed the project; W.W.Z. developed the catalytic asymmetric fluorinations; W.W.Z., X.Z. and J.K.A. conducted most of the experiments; J.Y.W., Y.P.C. and W.W.Z. conducted the DFT calculations and provided mechanism analysis; X.Y.M., J.K.L, Y.J.Q, H.Z. and J.J.X. helped the conduction of the project; and X.X.J. and W.W.Z. co-wrote the manuscript.

## Competing interests
The authors declare no competing interests.
