## [Peer Review File · Nature Communications]

REVIEWER COMMENTS

Reviewer #1 (Remarks to the Author):

The paper includes computational analysis of the reaction mechanism, however, I have several major reservations:

(1) the reaction proceeds at $-40\text{ }^{\circ}\text{C}$. This places an experimental upper bound on the activation barrier of around 18-19 kcal/mol. The computed energy surface has elementary steps with barriers of 24.3, 27.8 and 40.3 kcal/mol, none of which are feasible at this temperature.

(2) the paper doesn't clearly state what is the stereodetermining step, but it should be TS2 - in this regard, the relative energy of the diastereomeric structure for TS2 (i.e. giving the minor enantiomer) should be shown.

(3) Why is this particular stereochemistry preferred? The text mentions that the attack of F⁻ on the Si face is preferred, but no energies or structures are shown to support this.

(4) The barrier for Int 6 to Int 7 is impossibly high even at elevated temperatures.

(5) There are no methodological details about the calculations anywhere in the MS or SI (e.g. program, level of theory, basis set, solvation model) so it's impossible to know what has been done here. There are also no coordinates / absolute energies in the SI that should be provided as standard.

Overall, this section of the paper raises more questions than it addresses.

Reviewer #2 (Remarks to the Author):

In this manuscript, Jiang and co-workers report a catalytic asymmetric nucleophilic fluorination using $\text{BF}_3 \cdot \text{Et}_2\text{O}$, acting as fluorine source and dual-activating reagent. This method avoids the use of expensive any electrophilic fluorine reagent or high toxic and hazardous HF-based reagent, and the mechanism was reasonably proposed. However, the asymmetric catalyst system was quite similar with Jacobsen's work, and BF_3 as nucleophilic fluorine reagents was also not first reported. The publication of this paper on Nat Commun is recommended.

I have some minor comments:

1. It is preferred to cite representative works and related review. (For example, *J. Am. Chem. Soc.* 2020, 142, 14831–14837; *Chem. Rev.* 2015, 115, 566–611).
2. In Page 5, 7 the description "aminofluorination with N-cinnamylbenzamide" is not suitable.
3. In Fig 4, the dr values of some examples were relatively low. Could they be increased?
4. In Fig. 5, the potential nucleophilic fluorination process reported by Jacobsen's group through ArIF_2 should be further excluded through mechanism experiments.
5. In Fig. 6, the structures of TS1, Int3 and TS2 were wrong. The B-I bond should be B-O-I. This kind of careless mistakes should be avoided.
6. In Page 13, the description of s- NaHCO_3 was wrong. This kind of careless mistakes should be avoided.

Reviewer #3 (Remarks to the Author):

The authors present an enantioselective, oxidative fluorocyclization of allylic amides and related structures.

The reactions are well described and synthetically useful as they provide products of inherent value that are typically challenging to access.

The work is important and should, after appropriate revisions, be published. Whether Nature Communications is the right venue for this communication is not obvious given that there is very

limited conceptual novelty in what is reported here. That does not distract from the synthetic utility of the presented reaction but casts into question whether sufficient conceptual novelty is provided.

The authors describe an enantioselective cyclization reaction with a chiral aryl iodide with mCPBA as oxidant and a fluoride source as nucleophile, with an olefin bearing a nucleophile for intramolecular cyclization. That very same combination has been used by Jacobsen et al for example in JACS 2018, 140, 4797, with the same catalyst and the same oxidant. The novelty in what is reported here is a different alkene substrate class (endocyclic ring closure as opposed to exocyclic ring closure in the 2018 paper) and a different fluorine source BF₃ etherate is used here. The use of BF₃ etherate is the most relevant novelty that the authors mention several times. While BF₃ is possibly not an ideal fluoride source due to its strong Lewis acidity, I agree with the authors that, from a cost perspective, it represents a useful and desirable starting material for fluoride. I am ambivalent as to the BF₃ fluoride source constituting a sufficient conceptual advance for publication in Nature Communications.

I will state, however, that the products that are accessed are useful and they have not yet been easily accessible so far.

Before publication in any journal, the author must substantially revisit their writing as there are numerous inappropriate or just factually incorrect statements.

For example, the second sentence in the abstract: " However, catalytic asymmetric fluorination still remains elusive, " is just wrong, while "especially of the longstanding stereochemical challenges of the BF₃Et₂O-based fluorination have not yet been addressed" is at best confusing and awkward.

Response to the Comments of Reviewers in Detail

Reviewer #1:

The paper includes computational analysis of the reaction mechanism; however, I have several major reservations:

(1) The reaction proceeds at -40°C . This place an experimental upper bound on the activation barrier of around 18-19 kcal/mol. The computed energy surface has elementary steps with barriers of 24.3, 27.8 and 40.3 kcal/mol, none of which are feasible at this temperature.

Answer: Thank you very much for your significant questions. Your questions and suggestions are very important to improve our manuscript. We have re-conducted the DFT calculations with the help of a specialist in this area and we have given the revised picture of possible mechanism and DFT free energy profile in the revised manuscript (Fig. 6). In the revised calculated free energy profile, the energy barrier for the reaction between **Int2** ($\text{ArI}^+\text{OB}^-\text{F}_3$) and **1a** to generate **Int3** is only 3.6 kcal/mol. Besides, the energy barrier for the transformation from **Int3**⁺ to **TS1** is 13.1 kcal/mol and the energy barrier for the formation of **TS2** from **Int4**⁺ is 11.1 kcal/mol. In addition, based on the current DFT calculations results, we thought the key factors for the success of our reaction are: (1) the **Int1** (PhIO): since the control experiments indicate that our catalytic process (hypervalent iodine catalyst + $\text{BF}_3 \cdot \text{Et}_2\text{O}$) is different from Jacobsen's protocol (hypervalent iodine catalyst + $\text{Py} \cdot \text{HF}$), it might because the PhIO will not tend to directly oxidize alkenes as radical positive ions while some PhIL_2 -like oxidants are prone to undergo single electron transfer process and thus results in unpredictable reactions; (2) the interaction of PhIO and BF_3 could generate **Int2** ($\text{PhI}^+\text{-OB}^-\text{F}_3$), which enhanced the electrophilicity of the "hypervalent chiral iodine catalyst", however, PhIL_2 needs to "release" a ligand to produce similar species, this step is not necessarily beneficial in energy; (3) the leaving of OB^-F_3 group could generate $[\text{BF}_2\text{OB}^-\text{F}_3]^-$, forming two B-O bond, which may be more favorable to transform to $[\text{BF}_4]^-$ than PhIF_2 . Also, activation of substrates by Lewis acid may contribute to the formation of final fluorination products (supposed according to previous work, such as *Curr. Org. Chem.* **24**, 1263-1273 (2020); *J. Org. Chem.* **84**, 10402–10411 (2019) and *J. Am. Chem. Soc.* **131**, 5070–5071 (2009)).

(2) The paper doesn't clearly state what is the stereodetermining step, but it should be TS2 - in this regard, the relative energy of the diastereomeric structure for TS2 (i.e. giving the minor enantiomer) should be shown.

Answer: Thank you very much for your significant question. We thought there are two stereodetermining steps. The first step is the intermolecular nucleophilic attack of "F⁻" on the double bond, which produce **TS1**. In our revised manuscript, we give a calculated result in energy to demonstrate the more favorable transition state; For the impacts of steric hindrance, the second step is the intramolecularly nucleophilic attack of the "Ar¹ ring" on *Si* face at *C2* position, this is in accordance with the X-ray single crystal diffraction results.

(3) Why is this particular stereochemistry preferred? The text mentions that the attack of F⁻ on the *Si* face is preferred, but no energies or structures are shown to support this.

Answer: Thank you very much for your important question and advice. As seen from the energy profile in our revised manuscript (Fig. 6), the enantioselectivity is determined by the addition of F⁻ with [BF₄]⁻ as fluorine source to the cationic **Int3⁺**, the two **TSs** (**TS1** and **ent-TS1**) exhibit huge energy difference of 14.9 kcal/mol. A closer observation on the geometry shows that **ent-TS1** is much later, and bears a significantly longer I-O distance. Thus, the reported stereocontrol step is preferred.

(4) The barrier for **Int 6** to **Int 7** is impossibly high even at elevated temperatures.

Answer: Thank you very much. We have reconducted the DFT calculations with the help of a specialist in this area. The revised DFT calculated free energy profile were shown in Fig. 6 in the revised manuscript.

(5) There are no methodological details about the calculations anywhere in the MS or SI (e.g. program, level of theory, basis set, solvation model) so it's impossible to know what has been done here. There are also no coordinates / absolute energies in the SI that should be provided as standard.

Answer: Thank you very much for your important comments. We have added the methodological details about the calculations in the "Methods" section and supporting information (supporting information, section 8, pages S75-S90).

Reviewer #2:

1. It is preferred to cite representative works and related review. (For example, J. Am. Chem. Soc. 2020, 142, 14831–14837; Chem. Rev. 2015, 115, 566–611).

Answer: Thank you very much for your significant advice. We have cited the mentioned references in our revised manuscript, which were marked as highlight.

2. In Page 5, 7 the description “aminofluorination with *N*-cinnamylbenzamide” is not suitable.

Answer: Many thanks for your comment, we have modified the description as “aminofluorination of *N*-cinnamylbenzamide”, which was marked as highlight in our revised manuscript.

3. In Fig 4, the dr values of some examples were relatively low. Could they be increased?

Answer: Thank you very much. After reconduction of the catalytic process and checking the reaction results, we can find a tendency that the fluorinated products could be obtained with lower dr values when *N*-cinnamylbenzamides with electron withdrawing group on Ar² ring were applied to undergo the catalytic process. For examples, the **11b-13b** and **15b** displayed much lower dr values than **1b**, or other electron-donating group substituted substrates. Thus, the dr values of products bearing electron withdrawing group on Ar² ring could not be obviously increased in our catalytic system, we are sorry for that.

4. In Fig. 5, the potential nucleophilic fluorination process reported by Jacobsen’s group through ArIF₂ should be further excluded through mechanism experiments.

Answer: Thank you very much. According to your significant comment, we have done the control experiment using PhIF₂ as the “hypervalent iodine reagents” and “fluorine source” to evaluate the process. In fact, No desired product (**1b**) was observed, we have added this control experiment in Fig. 5, equation 1. This might because the PhIO will not tend to directly oxidize alkenes as radical positive ions while some

PhIL₂-like oxidants are prone to undergo single electron transfer process and thus results in unpredictable reactions.

5. In Fig. 6, the structures of **TS1**, **Int3** and **TS2** were wrong. The B-I bond should be B-O-I. This kind of careless mistakes should be avoided.

Answer: Thank you very much. We have reconducted the DFT calculations and give the revised calculated free energy profile in our revised manuscript (Fig. 6).

6. In Page 13, the description of s-NaHCO₃ was wrong. This kind of careless mistakes should be avoided.

Answer: Thank you very much. We have corrected our mistakes in the revised manuscript and marked as highlight. Also, we examined the whole manuscript carefully so that we can find the careless mistakes as much as possible and correct them. The revised sentences or words were marked as highlight in our revised manuscript.

Reviewer #3:

The work is important and should, after appropriate revisions, be published. Whether Nature Communications is the right venue for this communication is not obvious given that there is very limited conceptual novelty in what is reported here. That does not distract from the synthetic utility of the presented reaction but casts into question whether sufficient conceptual novelty is provided.

The authors describe an enantioselective cyclization reaction with a chiral aryl iodide with *m*-CPBA as oxidant and a fluoride source as nucleophile, with an olefin bearing a nucleophile for intramolecular cyclization. That very same combination has been used by Jacobsen et al for example in JACS 2018, 140, 4797, with the same catalyst and the same oxidant. The novelty in what is reported here is a different alkene substrate class (endocyclic ring closure as opposed to exocyclic ring closure in the 2018 paper) and a different fluorine source BF₃ etherate is used here. The use of BF₃ etherate is the most relevant novelty that the authors mention several times.

While BF_3 is possibly not an ideal fluoride source due to its strong Lewis acidity, I agree with the authors that, from a cost perspective, it represents a useful and desirable starting material for fluoride. I am ambivalent as to the BF_3 fluoride source constituting a sufficient conceptual advance for publication in Nature Communications.

I will state, however, that the products that are accessed are useful and they have not yet been easily accessible so far.

Before publication in any journal, the author must substantially revisit their writing as there are numerous inappropriate or just factually incorrect statements. For example, the second sentence in the abstract: " However, catalytic asymmetric fluorination still remains elusive, " is just wrong, while "especially of the longstanding stereochemical challenges of the $\text{BF}_3 \cdot \text{Et}_2\text{O}$ -based fluorination have not yet been addressed" is at best confusing and awkward).

Answer: Thank you very much for your significant advice. As we mentioned in our manuscript, the use of $\text{BF}_3 \cdot \text{Et}_2\text{O}$ as fluorine source in catalytic asymmetric fluorinations in the presence of chiral iodine catalyst is the most relevant novelty of our research, and we reported this protocol for the first time. Herein, $\text{BF}_3 \cdot \text{Et}_2\text{O}$ played multiple roles, for example, activation of the $\text{PhI}=\text{O}$, as the nucleophilic fluorine source. Besides, as you mentioned, $\text{BF}_3 \cdot \text{Et}_2\text{O}$ is a strong Lewis acid, so, the Lewis acid mediated Background reaction (see our reported research: *Current Organic Chemistry*, 2020, 24, 1263-1273) and hypervalent iodine catalyzed fluorination are competitive reactions. To our delight, by screening the reaction conditions, the desired fluorine products could be obtained through the current process.

We have carefully checked our manuscript and revised the incorrect sentences, for example, the statement "However, catalytic asymmetric fluorination still remains elusive, especially of the longstanding stereochemical challenges of the $\text{BF}_3 \cdot \text{Et}_2\text{O}$ -based fluorination have not yet been addressed" was modified as "However, catalytic asymmetric fluorination remains elusive, especially the longstanding stereochemical challenges which existed in $\text{BF}_3 \cdot \text{Et}_2\text{O}$ -based fluorinations, have not yet been addressed."; other modifications of the sentences were marked as highlight in our revised manuscript.

REVIEWERS' COMMENTS

Reviewer #1 (Remarks to the Author):

The authors present a revised DFT energy profile that is now consistent with the reaction temperature.

The revised text states that the competing transition structures display a "huge energy difference of 14.9 kcal/mol". Experimentally, the enantioselectivity for substrate 1b is 86% ee, while others can give higher selectivities. Taking the most selective reactions giving 99% ee, this corresponds to a DDG around 2.5 kcal/mol (at 0 degrees C). In light of this, the value of 14.9 kcal/mol does indeed seem rather at odds with the observations.

Reviewer 3 remarked that the claim "catalytic asymmetric fluorination still remains elusive" is wrong, however, the revision doesn't correct this.

Reviewer #2 (Remarks were to the Editor only)

This reviewer recommended publication of the paper.

Response to the Comments of Reviewers in Detail

Reviewer #1:

The authors present a revised DFT energy profile that is now consistent with the reaction temperature.

The revised text states that the competing transition structures display a "huge energy difference of 14.9 kcal/mol". Experimentally, the enantioselectivity for substrate **1b** is 86% ee, while others can give higher selectivities. Taking the most selective reactions giving 99% ee, this corresponds to a DDG around 2.5 kcal/mol (at 0°C). In light of this, the value of 14.9 kcal/mol does indeed seem rather at odds with the observations.

Answer: Thank you very much for your kind comments. Indeed, the Gibbs free energy change of 14.9 kcal/mol is surprising, which is much larger than a common observation in enantioselective reactions. Since the TSs involve some unique electronic structure and the combination of ionic species, which might be challenging for DFT calculation and implicit solvation model. We carefully performed an examination on the whole procedure, as well as a conformation search based on a molecular dynamics trajectory with the xtb program, in order to confirm the presented results. Also, the stability of wavefunction was examined, and no open shell ground state is involved. As a result, it is suggested that the level of theory employed in this work (B3LYP-D3BJ/6-311+G(d,p)/def2-TZVP/SDD) indeed gives a large energy difference.

Reviewer #2:

This reviewer recommended publication of the paper.

Answer: Thank you very much for your favorable consideration.

Reviewer #3:

The claim "catalytic asymmetric fluorination still remains elusive" is wrong, however, the revision doesn't correct this.

Answer: Thank you very much. The sentence was modified as "challenges in catalytic

asymmetric fluorinations, especially the longstanding stereochemical challenges existed in $\text{BF}_3 \cdot \text{Et}_2\text{O}$ -based fluorinations".